# Membranous nephropathy in the UK Biobank

**Patrick Hamilton**[1,2,3☯]*, **Kieran Blaikie**[1,4☯], **Stephen A. Roberts**[1,4], **Matthew Gittins**[1,4], **Mallory L. Downie**[5], **Sanjana Gupta**[5], **Catalin Voinescu**[5], **Durga Kanigicherla**[1,3], **Horia Stanescu**[5], **Robert Kleta**[5], **Paul Brenchley**[1,3]

**1** Manchester Academic Health Science Centre (MAHSC), The University of Manchester, Manchester, United Kingdom, **2** Faculty of Biology Medicine and Health, Division of Cell Matrix Biology and Regenerative Medicine, Wellcome Centre for Cell-Matrix Research, School of Biological Sciences, The University of Manchester, Manchester, United Kingdom, **3** Manchester Institute of Nephrology and Transplantation, Manchester Royal Infirmary, Manchester, United Kingdom, **4** Centre for Biostatistics, The University of Manchester, Manchester, United Kingdom, **5** UCL Department of Renal Medicine, UCL Medical School, London, United Kingdom

☯ These authors contributed equally to this work.

\* patrick.hamilton@manchester.ac.uk

## Abstract

### Background

Despite MN being one of the most common causes of nephrotic syndrome worldwide, its biological and environmental determinants are poorly understood in large-part due to it being a rare disease. Making use of the UK Biobank, a unique resource holding a clinical dataset and stored DNA, serum and urine for ~500,000 participants, this study aims to address this gap in understanding.

### Methods

The primary outcome was putative MN as defined by ICD-10 codes occurring in the UK Biobank. Univariate relative risk regression modelling was used to assess the associations between the incidence of MN and related phenotypes with sociodemographic, environmental exposures, and previously described increased-risk SNPs.

### Results

502,507 patients were included in the study of whom 100 were found to have a putative diagnosis of MN; 36 at baseline and 64 during the follow-up. Prevalence at baseline and last follow-up were 72 and 199 cases/million respectively. At baseline, as expected, the majority of those previously diagnosed with MN had proteinuria, and there was already evidence of proteinuria in patients diagnosed within the first 5 years of follow-up. The highest incidence rate for MN in patients was seen in those homozygous for the high-risk alleles (9.9/100,000 person-years).

### Conclusion

It is feasible to putatively identify patients with MN in the UK Biobank and cases are still accumulating. This study shows the chronicity of disease with proteinuria present years

**Data Availability Statement:** Data held by UK Biobank (http:/ukbiobank.org) - access to data is dependent on application directly to UK biobank All

data is from the UK biobank and therefore not permitted to be shared publicly. However, the data can be accessed by applying directly to the UK Biobank by any researcher worldwide through its standard application process - https://www.ukbiobank.ac.uk/enable-your-research/apply-for-access.

**Funding:** This was an approved study (I.D. 1618) by UK Biobank (http:/ukbiobank.org). We acknowledge funding from Kidney Research UK (KRUK) for the Stoneygate Foundation Grant JFS_IN_003_20160914 MLD was supported by the KRESCENT post-doctoral fellowship from the Kidney Foundation of Canada There was no additional external funding received for this study'.

**Competing interests:** The authors have declared that no competing interests exist

before diagnosis. Genetics plays an important role in disease pathogenesis, with the at-risk group providing a potential population for recall.

## Introduction

Membranous nephropathy (MN) is among the most common causes of adult nephrotic syndrome worldwide and has a significant healthcare burden, despite being a rare disease (10–12 cases per million) [1]. For the majority of patients, it is an autoimmune disease associated with the anti-phospholipase M-type receptor autoantibody (anti-PLA$_2$R); a highly sensitive biomarker and seemingly pathogenic factor in its own right [2,3]. There appears to be a strong genetic contribution to the condition, with the possession of single-nucleotide polymorphisms (SNPs) at *PLA2R1* and the Major Histocompatibility Complex, Class II, DQA1 (*HLA-DQA1*) conferring a dramatically increased risk of developing the disease [4–6]. The most recent GWAS has also shown two further loci associated with disease risk in *NKFB1* and *IRF4* [6]. However, given its rarity, understanding not only the disease pathogenesis but also the risk of disease onset has been challenging. Why some patients with the genetic predisposition develop a pathogenic antibody whilst others do not, has still not been elicited. Circumstantial evidence suggests there may be a role for the loss of tolerance. As with other autoimmune conditions, the presence of the disease associated risk SNPs are unlikely to be solely responsible for the evolution of the disease. An accompanying trigger, be it environmental or pathogenic, at the correct time, is needed to initiate the immune cascade leading to symptoms. The podocyte antigen peptide sequence shows similarities to a cell wall enzyme in certain commonly encountered pathogens such as *Clostridia* species [7]. One study from China suggests a potential environmental trigger by identifying an association between air pollution and MN compared to other autoimmune glomerulopathies [8,9].

The UK Biobank holds a clinical dataset and stored deoxyribonucleic acid (DNA), serum and urine for over 500,000 participants, aged between 40–70 years old, recruited from across the United Kingdom from 2007–2010. This age profile corresponds to the age of onset of MN with a mean of 54 years. DNA has been genotyped on a genome-wide microarray with subsequent imputation.

Further health record data has been linked through the Hospital Episode Statistics (HES) and cancer and mortality statistics [10]. Participant place of birth, residential area and occupation are all recorded, providing an invaluable resource for the study of environmental and occupational pollution exposures [11–14]. With the deep phenotyping of each participant, lifelong follow-up, genotyping and health record linkage, the UK Biobank provides a unique and powerful tool to help understand the epidemiology of rare diseases such as MN. An initial step in the use of the UK Biobank is its validation as a tool for accurately identifying patients with MN. Neither International Classification of Disease tenth edition (ICD-10) nor UK Read codes (a coding system for clinical terms used in the UK) have an explicit code for MN, but both coding systems have codes that will include MN plus potentially several other diagnoses.

Here, for the first time, we have used the UK Biobank to demonstrate its feasibility and to investigate the genesis of the rare kidney disease, MN.

The specific study aims were:

a. to provide a detailed description of the numbers of participants with confirmed or possible MN within the UK Biobank to inform the identification of participants for further phenotyping and sampling or recall for assessments

b. to determine the prevalence and incidence rates of MN in this population

c. to stratify the UK Biobank population by the genetic risk of MN

d. to assess the feasibility of linking environmental factors (infection and occupational exposure) to triggering of autoimmunity in this genetically susceptible cohort

## Methods

This is a cohort study in which the primary outcome was putative Membranous nephropathy (MN), defined as any primary or secondary HES inpatient diagnosis of ICD-10 N02.2, N03.2, N04.2, or N05.2 occurring before the UK Biobank HES data refresh in March 2019. The nominal diagnosis date was taken as the earliest such record. This did not include self-reported diagnoses at the time of UK Biobank entry, or clinical diagnoses not resulting in hospital admission, which are not fully available in the Biobank dataset.

Primary care attendance data in the UK is provided by multiple separate entities, with which UK Biobank is working to allow for data linkage. At present, a subset of 228,957 UK Biobank participants had clinical attendances in primary care [11–13]. For these records putative MN was defined as the equivalent Read codes version 2 (READ2) K0A22, K0A32, K021., K011, K016., K031. or READ3 codes K0A22, K021., K011., K016., K031.

Our descriptive study examined multiple exposures including sociodemographic characteristics and medical history records recorded at the time of UK Biobank entry, as well as hospital inpatient ICD-10 diagnoses and the Office of Population Censuses and Surveys Classification of Interventions and Procedures version 4 (OPCS-4) operative procedures occurring prior to the UK Biobank data as of 10th December 2019. Additional exposures examined included algorithmically derived end-stage kidney disease (ESKD), measures of area-level residential air pollution, and Standard Occupational Classification (SOC)-2000 occupation-linked workplace airborne exposures as derived from the SOC codes and the Airborne Chemical Job Exposure Matrix (ACE JEM) [15]. Both CKD and Kidney disease derived from Biobank fields f.41202 and f.41204. In addition to this, CKD defined using the ICD 10 code N18 –chronic kidney disease. Kidney Disease was defined using the ICD codes N0* and N1* which includes kidney disease but not limited to CKD. The source of each variable is provided in S1 Table.

Given coeliac disease and type 1 diabetes mellitus classically share the same HLA class II alleles as primary MN, namely HLA-DQ2.5 [16–18], we investigated the incidence of these conditions in the UK biobank in relation to identified MN cases.

Residential linked air pollution estimates of particulate matter air pollutants with aerodynamic diameters <10μm (PM10) and <2.5μm (PM2.5 or fine PM), and gaseous air pollutions (Nitrogen Dioxide ($NO_2$) and Nitrogen Oxide (NO)) were generated for 2010 using Land Use Regression (LUR) modelling as part of the European Study of Cohorts and Air Pollution Effects (ESCAPE, http://www.escapeproject.eu/). Additional estimates for Nitrogen Dioxide were estimated for the years 2005–2007 from EU-wide air pollution maps (resolution 100 metre x 100 metre) based in the LUR models, full details of the model and its performance can be found online [19]. All effect estimates related to associations with annual estimates of each air pollution were expressed per $\mu gm^{-3}$.

Where descriptive statistics have been produced, the median and interquartile range (IQR) have been reported for continuous measures and the proportion and percentages per category have been reported for categorical variables. Where data is missing, the number of cases with missing data is reported. In reporting disease incidence, the number of incident cases occurring post-UK Biobank entry has been divided by the total person-years of follow-up across the

disease-free UK Biobank population until March 1st 2019, accounting for individual date of UK Biobank entry and recorded date of death if applicable. Baseline patients defined as those with a diagnosis of MN on study entry, 'Early Incident' defined as those diagnosed within five years after study entry, and 'Late Incident' defined as those patients diagnosed later than 5 years after study entry.

SNPs associated with MN were identified as the lead independent SNPs from 3 previous GWAS studies, referred to as GWAS1-3. Allele counts identified in GWAS1[4] were obtained from the directly genotyped Biobank dataset. Allele counts from GWAS2[5] and GWAS3[6] were obtained from the imputed Biobank dataset. Imputed genotypes were used as called regardless of any quality assessment.

Associations between dichotomous phenotypes and genotypes were assessed by determining the relative risk (RR) associated with each genotype compared to homozygous reference alleles, and the RR per allele (additive genetic model).

To determine univariate association between exposures of interest and MN diagnosis, generalised linear models were used with non-cases as the reference category. Results were presented as relative risk ratios (RR) with accompanying 95% Wald confidence intervals.

All statistical analyses were performed in R Version 3.5.1 [20]. Plink v2.00 and BCFtools v1.10.2 were used to extract genetic data from the UK Biobank [21–24].

### Ethics statement

The UK Biobank has ethical approval from the North West Multi-centre Research Ethics Committee (MREC reference 21/NW0157) as a Research Tissue Bank (RTB) and therefore no separate ethical approval was required.

## Results

### MN incidence and prevalence

A total of 502,507 patients were included in the study. Based on hospital admissions, 100 participants were found to have a putative diagnosis of MN as indicated by the ICD 10 codes, 36 at baseline and 64 occurring during the follow-up period. At baseline, the prevalence of MN in the Biobank population was 72 cases per million, with 199 cases per million at the latest follow up. There was a total of 64 incident cases, with N052 contributing the largest number of patients at 31. With approximately 4.98 million total person-years of follow-up across the UK Biobank population, this corresponded to an incidence rate of 1.29 per 100,000 person-years, with a continued rise over time as the Biobank population ages. Table 1 and Fig 1.

### Demographics of MN cases

In the MN cohort, the median age at baseline assessment was 62 years old (IQR 56–65), compared to a median of 58 (IQR 50–63) in the non-MN group. There were 39% males in the MN cohort, compared to 54.4% males in the non-MN cohort. The majority of participants in the UK Biobank with and without MN were white, with similar BMI, smoking status, alcohol consumption and levels of deprivation as measured by the Townsend Deprivation Index and by the Index of Multiple Deprivation Table 2.

### Clinical

There was a higher level of proteinuria at baseline in the MN group compared to the non-MN group, with a median urinary albumin:creatinine ratio (uACR) of 3.0 mg/mmol (IQR 0.4–22.9) and 0.4 mg/mmol (IQR 0.4–0.6) respectively. Kidney function at baseline was also lower

**Table 1. Incidence and prevalence.**

| ICD10 | Description | Cases Detected | | Prevalence per million | | Incident Cases | Incidence Rate (per 100,000 person-years) |
|---|---|---|---|---|---|---|---|
| | | Baseline | Last follow up | Baseline | Last follow up | | |
| N022 | Recurrent and persistent haematuria with diffuse membranous glomerulonephritis | 5 | 35 | 10 | 70 | 30 | 0.60 |
| N032 | Chronic nephritic syndrome with diffuse membranous glomerulonephritis | 4 | 12 | 8 | 24 | 8 | 0.16 |
| N042 | Nephrotic syndrome with diffuse membranous glomerulonephritis | 4 | 27 | 8 | 54 | 23 | 0.46 |
| N052 | Unspecified nephritic syndrome with diffuse membranous glomerulonephritis | 25 | 56 | 50 | 111 | 31 | 0.62 |
| Total | Putative Membranous Nephropathy | 36 | 100 | 72 | 199 | 64 | 1.29 |

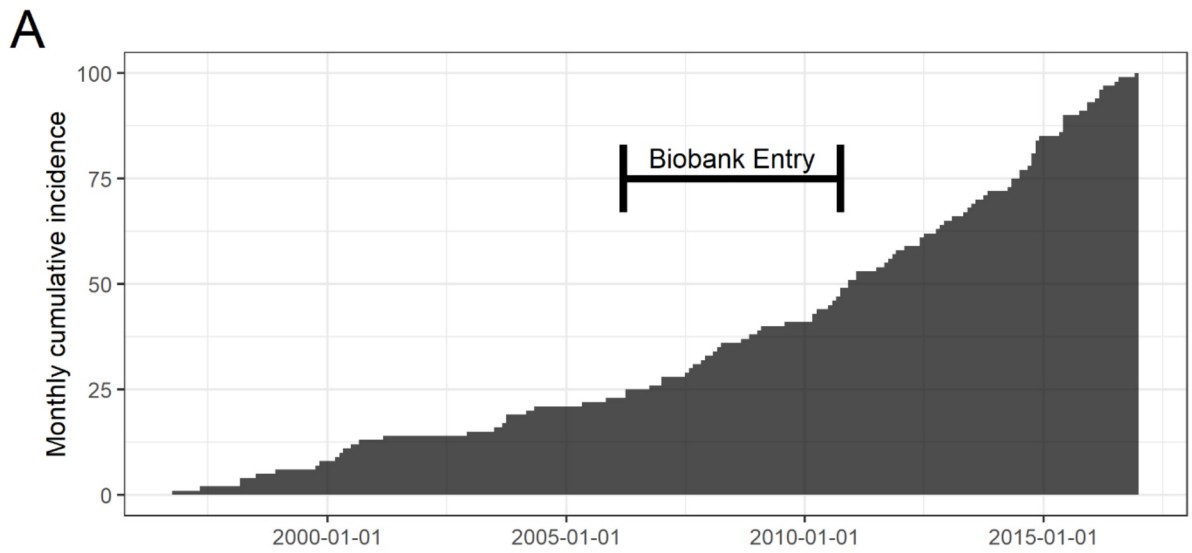

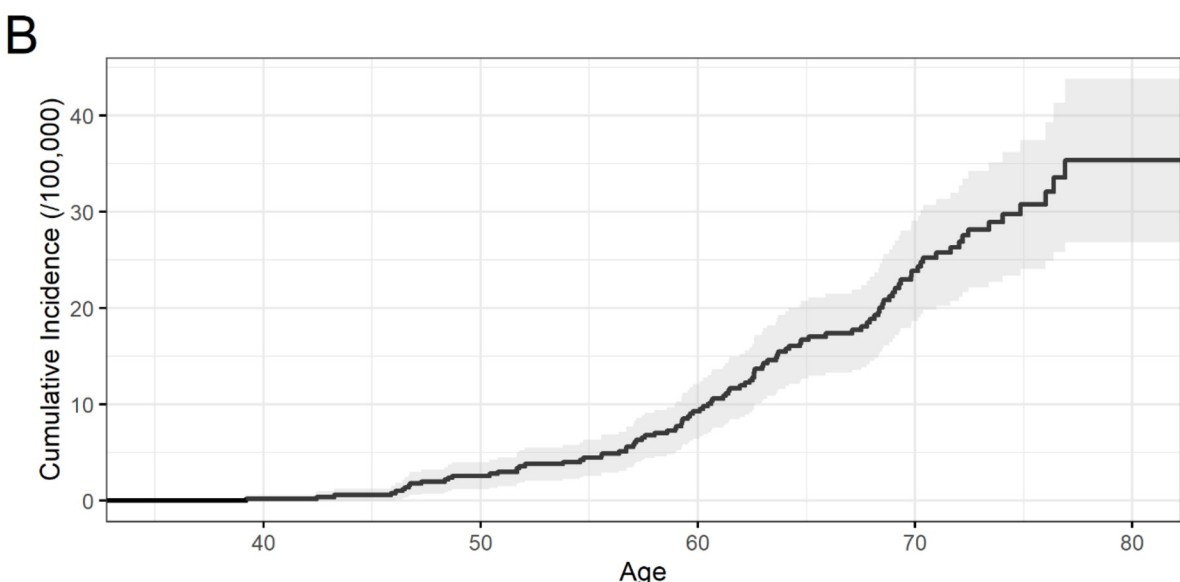

**Fig 1. Monthly cumulative incidence of MN.** A) Cases identified each calendar month B) Kaplan-Meier estimate by age with 95%CI.

**Table 2. Demographics for the UK Biobank population.**

| Characteristics | | No MN | MN |
|---|---|---|---|
| *N* | | 502407 | 100 |
| Age at baseline assessment | Median [IQR] | 58 [50–63] | 62 [56–65] |
| Age at last follow-up (death or end of HES data) | Median [IQR] | 68 [60–73] | 70 (65–74) |
| Sex | Male | 273345/502407 (54.4%) | 39/100 (39%) |
| | Female | 229062/502407 (45.6%) | 61/100 (61%) |
| Ethnicity | White | 472608/501509 (94.2%) | 90/100 (90%) |
| | Asian | 9877/501509 (2.0%) | 5/100 (5%) |
| | Black | 8059/501509 (1.6%) | 2/100 (2%) |
| | Mixed/Other | 9087/501509 (1.8%) | 3/100 (3%) |
| | Not Known | 1878/501509 (0.4%) | 0/100 (0%) |
| BMI | Median [IQR] | 26.7 [24.1–29.9] | 27.5 [24.5–30.7] |
| | NA | 3104 | 1 |
| Smoking Status | Prefer not to answer | 2057/501516 (0.4%) | 0/100 (0%) |
| | Never | 273472/501516 (54.5%) | 50/100 (50%) |
| | Previous | 173018/501516 (34.5%) | 40/100 (40%) |
| | Current | 52969/501516 (10.6%) | 10/100 (10%) |
| Alcohol Consumption | Prefer not to answer | 756/501510 (0.2%) | 0/100 (0%) |
| | Never | 22377/501510 (4.5%) | 8/100 (8%) |
| | Previous | 18102/501510 (3.6%) | 2/100 (2%) |
| | Current | 460275/501510 (91.8%) | 90/100 (90%) |
| Alcohol Consumption Frequency | Prefer not to answer | 604/501510 (0.1%) | 0/100 (0%) |
| | Daily or almost daily | 101751/501510 (20.3%) | 18/100 (18%) |
| | Three or four times a week | 115422/501510 (23.0%) | 18/100 (18%) |
| | Once or twice a week | 129265/501510 (25.8%) | 27/100 (27%) |
| | One to three times a month | 55848/501510 (11.1%) | 7/100 (7%) |
| | Special occasions | 57989/501510 (11.6%) | 20/100 (20%) |
| | Never | 40631/501510 (8.1%) | 10/100 (10%) |
| Townsend Deprivation Index | Median [IQR] | -2.1 [-3.6–0.6] | -1.9 [-3.9–0.5] |
| | NA | 623 | 0 |
| Index of Multiple Deprivation | Median [IQR] | 12.8 [7.1–23.3] | 16.5 [8.4–27.6] |
| | NA | 12737 | 0 |

in the MN group with a median estimated glomerular filtration rate (eGFR) 78.1ml/min/1.73m$^2$ (IQR 52.2–93.6), compared to the non-MN cohort who had a median eGFR 92.8ml/min/1.73m$^2$ (IQR 82.9–100.1) Table 3.

For those diagnosed with MN prior to study recruitment, a majority (66%; n = 21) had proteinuria at baseline with a uACR of more than 3mg/mmol. 28% (n = 9) had moderately increased albuminuria with a uACR of more than 30mg/mmol. For those diagnosed within 5

**Table 3. Phenotype per group.**

| | | No MN | MN | | | |
|---|---|---|---|---|---|---|
| | | | **Any** | **Baseline** | **Early Incident** | **Late Incident** |
| | | **502,407** | **100** | **36** | **36** | **28** |
| Self-reported Kidney Failure | | 872/502407 (0.2%) | 12/100 (12%) | 8/36 (22%) | 4/36 (11%) | 0/28 (0%) |
| HES CKD | | 9033/502407 (1.8%) | 49/100 (49%) | 16/36 (44.%) | 20/36 (56%) | 13/28 (46%) |
| HES Kidney Disease | | 21931/502407 (4.4%) | 100/100 (100%) | 36/36 (100%) | 36/36 (100%) | 28/28 (100%) |
| Algorithmically derived ESKD | | 1168/502407 (0.2%) | 23/100 (23%) | 8/36 (22%) | 11/36 (31%) | 4/28 (14%) |
| uACR at baseline (mg/mmol) | <3 | 458536/484201 (94.7%) | 47/93 (50.5%) | 11/32 (34%) | 18/35 (51%) | 18/26 (69%) |
| | 3–30 | 23405/484201 (4.8%) | 23/93 (25%) | 12/32 (38%) | 7/35 (20%) | 4/26 (15%) |
| | 30–250 | 2128/484201 (0.4%) | 15/93 (16%) | 7/32 (22%) | 4/35 (11%) | 4/26 (15%) |
| | >250 | 132/484201 (0.03%) | 8/93 (8.6%) | 2/32 (6%) | 6/35 (17%) | 0/26 (0%) |
| | Median (IQR) | 0.4 (0.4–0.6) | 3.0 (0.4–22.9) | 7.2 (2.6–31.2) | 1.9 (0.4–99.2) | 0.6 (0.4–8.8) |
| | Unknown | 18206 | 7 | 4 | 1 | 2 |
| eGFR at baseline (ml/min/1.73m$^2$) | ≥90 | 277341/468781 (59.2%) | 32/89 (36%) | 6/31 (19.4%) | 11/33 (33%) | 15/25 (60.0%) |
| | 60–90 | 180729/468781 (38.6%) | 31/89 (35%) | 13/31 (42%) | 10/33 (30%) | 8/25 (32%) |
| | 15–60 | 10571/468781 (2.3%) | 24/89 (27%) | 11/31 (36%) | 11/33 (33%) | 2/25 (8%) |
| | <15 | 140/468781 (0.03%) | 2/89 (2%) | 1/31 (3%) | 1/33 (3.0%) | 0/25 (0%) |
| | Median (IQR) | 92.8 (82.9–100.1) | 78.1 (52.2–93.6) | 74.3 (46.8–86.7) | 76.8 (48.8–93.8) | 92.9 (76.3–95.0) |
| | Unknown | 33626 | 11 | 5 | 3 | 3 |
| Self-reported Diabetes | | 25468/502407 (5.1%) | 12/100 (12%) | 5/36 (14%) | 5/36 (14%) | 2/28 (7.%) |
| HES Diabetes | | 31266/502407 (6.2%) | 17/100 (17%) | 8/36 (22%) | 6/36 (17%) | 3/28 (11%) |
| Self-reported Coeliac Disease | | 2049/502407 (0.4%) | 0/100 (0%) | 0/36 (0.) | 0/36 (0%) | 0/28 (0%) |
| HES Coeliac Disease | | 2481/502407 (0.5%) | 1/100 (1%) | 1/36 (3%) | 0/36 (0%) | 0/28 (0%) |
| Self-reported Kidney Biopsy | | 129/502407 (0.03%) | 3/100 (3%) | 3/36 (8%) | 0/36 (0%) | 0/28 (0%) |
| OPCS Kidney Biopsy | | 288/413517 (0.07%) | 6/100 (6%) | 3/36 (8%) | 2/36 (6%) | 1/28 (4%) |

Relationship between baseline cases, all incident cases, and incident cases < = 5 years from baseline.

MN = Membranous Nephropathy

HES = Hospital Episode Statistics

CKD = Chronic Kidney Disease

ESKD = End-Stage Kidney Disease

ACR = urinary albumin:creatinine ratio

eGFR = estimated Glomerular Filtration Rate

IQR = Inter-Quartile Range

SD = Standard Deviation

OPCS = OPCS Classification of Interventions and Procedures. See S1 Table for biobank variable codes and sources. eGFR calculated at baseline, algorithmically derived ESKD assessed throughout follow up period.

years of recruitment (early incident), there was already evidence of proteinuria at baseline, with 49% (n = 17) having a uACR of more than 3mg/mmol, and 29% (n = 10) with a uACR of more than 30mg/mmol. In the late incident group, diagnosed more than 5 years after recruitment, the majority had no evidence of proteinuria at baseline with 69% (n = 18) having a uACR of less than 3mg/mmol. For this late incident group there was proteinuria noted at baseline in 31% (n = 8) of patients (uACR greater than 3mg/mmol), and moderately increased albuminuria in 15% (n = 4) with a uACR of more than 30mg/mmol Table 3.

Kidney function for patients diagnosed prior to recruitment and in the early incident group, as measured by eGFR, was similar at baseline; median 74.3ml/min/1.73m$^2$ (IQR 46.8–86.7) in the baseline group, and median 76.8ml/min/1.73m$^2$ (IQR 48.8–93.8) in the early

incident group. In the late incident group, the median baseline eGFR was 92.9ml/min/1.73m$^2$ (IQR 76.3–95.0) Table 3.

## Genetics

In the total population, we found that 73.4% (n = 357,516) were homozygous for the low-risk *HLA-DQA1* alleles (CC) and 2.4% (n = 11,622) patients were homozygous for the high-risk allele (TT). For *PLA2R1*, 35.8% (n = 174,588) were homozygous for the high-risk alleles (AA) compared to 16.8% (n = 81,861) homozygous for the low-risk GG alleles. Of these, 0.8% (n = 4079) were found to have both high-risk allele SNPs. S2 and S3 Tables.

We see similar results in the UK Biobank population as in the original GWAS with respect to risk of MN diagnosis. Compared to those homozygous for the low-risk *HLADQ* allele (C), being homozygous for the high-risk allele (T) was associated with an 8.79 times greater risk of MN diagnosis (RR: 9.79, 95% CI: 5.36–17.85). For *PLA2R1*, compared to those homozygous for the low-risk allele (G), being homozygous for the high-risk allele (A) was associated with a 1.22 times greater risk of MN (RR: 2.22, 95% CI: 1.16–4.25). Considering *HLADQ* and *PLA2R1* in combination, using the low-risk allele combination CCGG as the reference, the relative risk of MN for those homozygous for both SNPs was 23.44 (95% CI: 7.67–71.62). There was no increased risk of MN associated with being homozygous for either of the two novel lead SNPs identified in the most recent GWAS[6]. For *NFKB1*, the relative risk of MN among those homozygous for the high-risk allele was 1.08 (95% CI: 0.56–2.10); and for *IRF4*, the RR among those homozygous for the high-risk allele was 1.72 (95% CI: 0.62–4.75). Table 4 and Fig 2.

The highest incidence rate of MN was seen in those homozygous for the high-risk *HLA-DQA1* allele (TT) and high-risk *PLA2R1* allele (AA), with an incidence rate of 9.9 cases per 100,000 person-years. The lowest-risk group (CCGG) had an incidence rate of 0.5 cases per 100,000 person-years Table 5.

Subgroup analysis comparing the cohorts homozygous for both the high-risk alleles for *HLA-DQA1* and *PLA2R1* (TT and AA respectively) with those not homozygous for both show a similar proportion of diabetes, both self-reported and through HES linkage. In the high-risk group, 2.8% of individuals self-reported having coeliac disease, and 3.2% were identified as having coeliac disease through HES linkage, this is compared to 0.4% and 0.5% respectively, in the low-risk group S4 Table.

At baseline, there was no difference in eGFR in patients homozygous for the high-risk *HLA-DQA1* and *PLA2R1* alleles (TTAA) compared to those not. For uACR, the high-risk group showed weak association with a higher degree of proteinuria over 30 mg/mmol; 0.71% among those homozygous for TTAA, compared to 0.47% in those not homozygous for the high-risk alleles (not TTAA) S4 Table.

Using patients with baseline proteinuria data, and homozygous for the low-risk allele for *HLA-DQA1* as reference, there was a weak negative association with moderately or severely increased albuminuria for those homozygous for the high-risk alleles (TT); (RR: 0.94, 95% CI: 0.86–1.02). Patients homozygous for the high-risk *PLA2R1* allele (AA) were slightly more likely to have moderately increased albuminaemia (RR: 1.05, 95% CI: 1.01–1.08) and severely increased albuminaemia (RR: 1.11, 95% CI: 0.99–1.26), compared to the low-risk allele group (CC) S5 Table.

## Pollution and work-related environmental exposures

Univariate comparisons indicated no significant associations between increased environmental exposure to Nitrogen Dioxide, Nitrogen Oxides, or Particulate Matter and risk of MN

**Table 4. Genetic risk analysis in the membranous nephropathy cohort using the lead SNPs from GWAS1[4], GWAS2[5] and GWAS3[6] from the imputed UK Biobank dataset.**

| | | | MN | RR (95% CI) | GWAS OR |
|---|---|---|---|---|---|
| GWAS1[4] | HLADQA1 rs2187668 | CC | 44/357516 (0.01%) | 1 | |
| | | CT | 39/118070 (0.03%) | 2.68 (1.74–4.13) | |
| | | TT | 14/11622 (0.12%) | 9.79 (5.36–17.85) | |
| | | OR Per allele | | 3.11 (2.31–4.18) | 4.32 (3.73–5.01) |
| | PLA2R1 rs4664308 | GG | 11/81861 (0.01%) | 1 | |
| | | AG | 34/230759 (0.02%) | 1.10 (0.56–2.16) | |
| | | AA | 52/174588 (0.03%) | 2.22 (1.16–4.25) | |
| | | OR Per allele | | 1.66 (1.22–2.26) | 2.28 (1.9–2.64) |
| GWAS2[5] | HLADQA1 rs9272729 | GG | 44/351417 (0.01%) | 1 | |
| | | AG | 41/117731 (0.04%) | 2.78 (1.82–4.26) | |
| | | AA | 12/10747 (0.11%) | 8.92 (4.71–16.88) | |
| | | OR Per allele | | 2.99 (2.17–4.11) | 6.86 (4.55–10.3) |
| | PLA2R1 rs17830558 | GG | 22/138318 (0.02%) | 1 | |
| | | TG | 40/228842 (0.02%) | 1.10 (0.65–1.85) | |
| | | TT | 33/105620 (0.03%) | 1.96 (1.15–3.37) | |
| | | OR Per allele | | 1.44 (1.08–1.90) | 1.63 (1.23–2.15) |
| GWAS3[6] | HLADRB1/DQA1 rs9271573 | CC | 24/170192 (0.01%) | 1 | |
| | | AC | 41/234411 (0.02%) | 1.24 (0.75–2.05) | |
| | | AA | 32/82375 (0.04%) | 2.75 (1.62–4.68) | |
| | | OR per allele | | 1.70 (1.28–2.25) | 2.41 (2.26–2.57) |
| | PLA2R1 rs17831251 | TT | 11/80771 (0.014%) | 1 | |
| | | CT | 33/227843 (0.01%) | 1.06 (0.54–2.10) | |
| | | CC | 51/172495 (0.03%) | 2.17 (1.13–4.16) | |
| | | OR per allele | | 1.67 (1.22–2.28) | 2.25 (2.09–2.42) |
| | NFKB1 rs230540 | CC | 11/59492 (0.02%) | 1 | |
| | | CT | 43/216576 (0.02%) | 1.07 (0.55–2.08) | |
| | | TT | 41/205265 (0.02%) | 1.08 (0.56–2.10) | |
| | | OR per allele | | 0.97 (0.72–1.31) | 1.25 (1.17–1.33) |
| | IRF4 rs9405192 | AA | 4/33043 (0.01%) | 1 | |
| | | GA | 39/182247 (0.02%) | 1.77 (0.63–4.95) | |
| | | GG | 54/259096 (0.02%) | 1.72 (0.62–4.75) | |
| | | OR per allele | | 0.90 (0.65–1.25) | 1.29 (1.21–1.37) |
| GWAS1[4] SNP Combinations | HLADQA1 rs2187668 PLA2R1 rs4664308 | CCGG | 5/59756 (0.01%) | 1 | |
| | | CCAG | 17/168473 (0.01%) | 1.21 (0.44–3.27) | |
| | | CCAA | 22/129287 (0.02%) | 2.03 (0.77–5.37) | |
| | | CTGG | 4/20131 (0.02%) | 2.37 (0.64–8.84) | |
| | | CTAG | 13/56717 (0.02%) | 2.74 (0.98–7.68) | |
| | | CTAA | 22/41222 (0.05%) | 6.38 (2.42–16.84) | |
| | | TTAG | 4/5569 (0.07%) | 8.58 (2.31–31.96) | |
| | | TTGG | 2/1974 (0.10%) | 12.11 (2.35–62.37) | |
| | | TTAA | 8/4079 (0.20%) | 23.44 (7.67–71.62) | |
| GWAS1[4] Risk Groups | Low Risk | CCGG | 5/59756 (0.01%) | 1 | |
| | Medium Risk | All else | 84/423373 (0.02%) | 2.37 (0.96–5.85) | |
| | High Risk | TTAA | 8/4079 (0.20%) | 23.44 (7.67–71.62) | |

*(Continued)*

**Table 4.** (Continued)

| | | | MN | RR (95% CI) | GWAS OR |
|---|---|---|---|---|---|
| GWAS1[4] PLA2R1 Groups | No PLA2R1 allele | CCGG, CTGG, TTGG | 11/81861 (0.01%) | 1 | |
| | PLA2R1 Allele | All Else | 86/405347 (0.02%) | 1.58 (0.84–2.96) | |

MN—Membranous Nephropathy

RR—Relative Risk

CI—Confidence Interval

GWAS—Genome-wide Association Study

OR—Odds Ratio

SNP—Single Nucleotide Polymorphism.

diagnosis. Likewise, no significant associations were found between work-related environmental exposures and risk of MN diagnosis, though comparisons were subject to small sample sizes among those exposed with MN Tables 6 and S6.

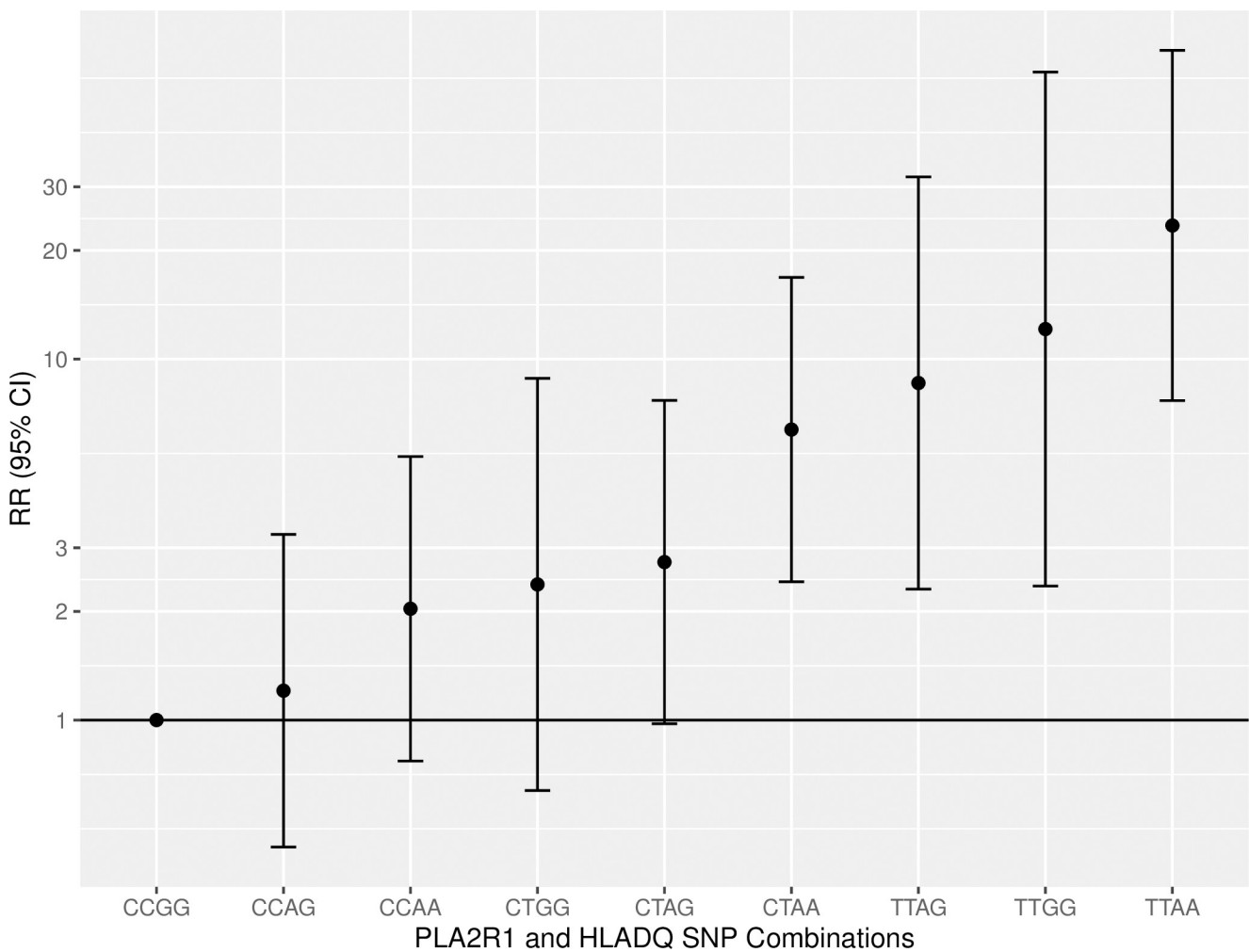

**Fig 2. Genetic Risk of MN by PLA2R1 and HLADQ.** Based on directly called SNPs determined in GWAS1[4].

**Table 5. Prevalence and incidence measures for: Those in high-risk and low-risk groups; those with a PLA2R1 allele and without a PLA2R1 allele.**

| Genotype Groups | | MN Prevalence | | Incident Cases | Total Follow-up (in 100,000 years) | Incidence Rate (per 100,000 years) |
|---|---|---|---|---|---|---|
| | | Baseline | Last follow up | | | |
| GWAS1 HLADQA1 and PLA2R1 SNP Combinations | CCGG | 2 | 5 | 3 | 5.92 | 0.5 |
| | CTGG | 1 | 4 | 3 | 2.00 | 1.5 |
| | TTGG | 0 | 2 | 2 | 2.00 | 10.2 |
| | CCAG | 5 | 17 | 12 | 16.68 | 0.7 |
| | CTAG | 5 | 13 | 8 | 5.62 | 1.4 |
| | TTAG | 1 | 4 | 3 | 0.55 | 5.5 |
| | CCAA | 4 | 22 | 18 | 12.77 | 1.4 |
| | CTAA | 13 | 22 | 9 | 4.08 | 2.2 |
| | TTAA | 4 | 8 | 4 | 0.40 | 9.9 |
| | Unknown HLADQ or PLA2R1 | 1 | 3 | 2 | 1.60 | 1.3 |
| Risk Groups | Low Risk (CCGG) | 2 | 5 | 3 | 5.92 | 0.5 |
| | Medium Risk (All Else) | 29 | 84 | 55 | 41.88 | 1.3 |
| | High Risk (TTAA) | 4 | 8 | 4 | 0.40 | 9.9 |
| PLA2R1 Allele Groups | No PLA2R1 Allele (CCGG, CTGG, TTGG) | 3 | 11 | 8 | 8.11 | 1.0 |
| | PLA2R1 Allele (All Else) | 32 | 86 | 54 | 40.10 | 1.3 |

## Primary care data

Data from primary care records was available for a subset of 228,957 (45.6%) participants. Of these, there were 38 patients identified as putative MN cases using the previously defined READ codes. Using hospital admissions data, 41 patients in this subset were identified as having MN using the previously defined ICD codes. Of all identified patients with MN, 19 were identified in both sources, 19 only in primary care and 22 only through hospital admissions. For those patients identified in both sources, the majority were coded in both databases at the same time, but there were a number of patients with diagnoses in the primary care database for whom diagnosis was coded years in advance of hospital admission. The age distribution at first hospital admission was similar among patients included and not included in the primary care database. However, some of the primary care cases not identified by hospital admissions were diagnosed at a younger age S1 Fig.

## Discussion

This is the first study to examine the occurrence and determinants of MN within the UK Biobank cohort. We have shown that putative MN cases can be identified at entry to the UK

**Table 6. Associations with environmental exposures presented in μgm⁻³: Pollution exposure: Identified MN cases versus controls with no diagnosed MN.**

| Pollution Metric | | No MN | MN | RR (95% CI) |
|---|---|---|---|---|
| | | Median [IQR] | Median [IQR] | |
| Nitrogen Dioxide | 2005 | 28.4 [23.1–34.7] | 28.9 [24.2–36.6] | 1.01 (0.99–1.03) |
| | 2006 | 27.9 [22.8–33.3] | 28.4 [23.7–35.2] | 1.01 (0.99–1.03) |
| | 2007 | 28.9 [23.6–35.0] | 30.3 [24.5–36.7] | 1.01 (0.99–1.03) |
| | 2010 | 26.2 [21.5–31.3] | 26.4 [21.8–30.9] | 1.01 (0.98–1.03) |
| Nitrogen Oxides | 2010 | 42.4 [34.4–50.8] | 42.9 [34.6–51.2] | 1.00 (0.99–1.01) |
| Particulate Matter | <2.5μm | 9.9 [9.3–10.6] | 10.0 [9.5–10.5] | 1.02 (0.84–1.23) |
| | <10μm | 16.0 [15.2–17.0] | 16.1 [15.4–16.8] | 0.98 (0.88–1.09) |
| Distance from home to main road (per 100 metres) | | 377.4 [166.9–751.9] | 378.8 [228.6–711.8] | 0.98 [0.95–1.02] |

Biobank study as well as throughout follow-up. The incidence rate calculated here of 1.29 diagnoses per 100,000 person-years is similar to previously reported rates in other studies [1,8,9,25–27]. Most previous studies reporting incidence rates for MN are based on retrospective data from biopsies–a possibly unrepresentative cohort given the inherent suspicion of an underlying kidney disease. In contrast, the UK Biobank cohort is ostensibly a population-based sample (see Fry *et al* for consideration of the sample representativeness [14]), with this study being the first to evaluate MN occurrence in such a population.

At present MN remains a biopsy-diagnosed condition requiring specialist input, usually in a tertiary referral centre in the UK. The coding for MN in primary care records is therefore generally based on correspondence from a tertiary care centre, meaning patients assigned codes for MN are highly likely to have the condition (unpublished work). However, there is no unifying code for MN in either ICD-10 or READ codes, and those that are available include adjunctive diagnoses. This raises the possibility that patients with MN could be missed, as they have been coded with more generic descriptions of the condition such as nephrotic syndrome. A number of patients identified from primary care data were diagnosed prior to the initiation of HES and who do not, therefore, appear in the hospital admissions database at all, or who appear years later. It may be that some of these patients were already in remission before the commencement of HES data, and therefore were never captured. At present, the available HES data also does not capture outpatient episodes, meaning patients already diagnosed prior to the commencement of the database, will not be identified as they are only seen in the outpatient clinic setting and not as an inpatient. Patients in remission could appear in the HES database years after identification in the General Practitioner (GP) database, owing to a relapse necessitating admission to a specialist nephrology centre. An ongoing limitation with data linkage in the UK Biobank is the incomplete coverage of both primary care data and HES, although this is being continuously addressed and updated.

MN is a disease of middle to late age, with the median age at baseline here of 62 years old [28,29]. We found a female predominance for MN in the UK Biobank, which is in-keeping with autoimmune disease in general although not specifically in MN [30]. A number of studies in the past have shown the opposite is true with MN, however, though this prior work has been inconsistent, emblematic of the difficulty of epidemiological studies in examining this rare disease [1].

As expected, there was a higher degree of proteinuria and lower eGFR at baseline in our study in patients with MN compared to the non-MN cohort. Interestingly, those patients who had no diagnosis of MN at baseline but who went on to be diagnosed within 5 years already had evidence of subclinical biochemical abnormalities suggestive of MN. Almost half of these early incident patients had some degree of proteinuria and over a third had an eGFR of less than 60ml/min/1.73m$^2$. These findings are similar to other recent studies [31], and are indicative of the chronicity and slowly progressive nature of the disease, with patients experiencing ongoing pathogenic autoantibody production and glomerular damage years before it becomes clinically apparent. Identification of this prodromal cohort in the UK Biobank offers a unique opportunity to confirm the early role of anti-PLA$_2$R in pathogenesis up to 5 years prior to clinical diagnosis of MN using the sera stored at study entry. It also allows testing of the hypothesis that infection could be the environmental trigger for MN by interrogation of the IgG antibody proteome library in early serum samples first positive for anti-PLA$_2$R, in comparison to age/gender-matched controls. It may be possible to identify a unique IgG phenotype to infectious agents coincident with the onset of anti-PLA$_2$R autoantibodies.

Currently 80% of autoimmune MN patients present with nephrotic syndrome without any prior indication of an underlying kidney disease, but with proteinuria and autoantibody production that may have been active for a variable period before this. Although the exact cause is

still unknown, knowledge of the disease pathogenesis has increased markedly since the discovery of the anti-PLA$_2$R autoantibody. There is undoubtedly a strong genetic component to autoimmune MN, with multiple GWAS initially identifying two genes, *HLA-DQA1* and *PLA2R1* in Europeans, and more recently *NFKB1* and *IRF4*, that all account for susceptibility to MN. Possession of homozygous pathological alleles of *HLA-DQA1* and *PLA2R1* raises the odds ratio from 1 to 78 for having the disease [4]. What is not known, is how likely it is for a patient to develop MN in the presence of these high-risk alleles. What is the lifetime risk of developing MN conferred by the high-risk alleles? Certainly, in our study, the relative risk of MN is significantly higher in patients homozygous for the *HLA-DQA1* and *PLA2R1* compared to those without any of the risk alleles (RR: 23.4, 95% CI: 7.7–71.6). Homozygosity of risk alleles in *HLA-DQA1* and *PLA2R1* resulted in a higher incidence of MN (9.9 cases per 100,000 person-years) and a higher degree of macro-albuminuria. This suggests not only a higher incidence but also a more severe phenotype in keeping with previous work showing an association of individuals with the *HLA-DQA1* risk alleles having higher anti-PLA2R antibodies [29]. There was a low penetrance with only 0.20% of patients homozygous for the high-risk alleles having, or developing MN, during the study follow-up, but as UK Biobank is a lifetime epidemiology study, the risk of developing MN will become evident. Given the chronicity of MN disease and its diagnosis late in life for most patients, this is a group of patients in whom active follow-up would be essential in order to determine the strength of genetics on the development of the disease.

We found no evidence for an increase in MN diagnosis in relation to particulate exposure or occupational exposures to heavy metals and hydrocarbons. This may be related to the much lower levels of exposure among patients in the UK compared to China, where this association was first described in biopsy patients [8,9]. In our cohort, the median level of exposure over the study period was 10.0μgm$^{-3}$ for particulate matter of less than 2.5μm, compared to a mean of 55.6μgm$^{-3}$ (range 8.1 to 110.5) in China. Limitations to this comparison include the small numbers of exposed cases, lack of data on previous exposures, and the study population being generally more educated and from a higher socioeconomic background compared to the general population, and therefore potentially less likely to be exposed to higher pollution levels–all influencing the statistical power of environmental comparisons and our ability to detect plausible magnitude associations. A significantly larger study population would be required to investigate this further at lower particulate matter levels, and at present would be outside the remit of the UK Biobank.

## Conclusion

Here we have shown for the first time that it is feasible to putatively identify patients with MN in the UK Biobank, both at study entry and over time, allowing for the use of this powerful resource to study initiation of a rare autoimmune disease in greater depth than previously possible, however definitive diagnosis would require participant or stored sample access. This study provides further evidence for the chronicity of disease with proteinuria present years prior to diagnosis in a number of patients. Genetics also plays an important role in the development of the disease although with low penetrance. Patients with the high-risk alleles provide an invaluable study population for prospective investigation of the disease process, this is particularly pertinent given the potential for missed cases in the database. This at-risk group would be an important sample for recall to interrogate further. Despite the low numbers of MN patients identified, this is the most comprehensive study of MN in a defined population to date. Importantly it also provides a baseline for further work to understand the pathogenesis of the disease as follow-up progresses.

## Supporting information

**S1 Fig. A) Time difference from GP diagnosis to HES diagnosis where the case was identified in both sources.** B) Age distributions of the cases identified in the two sources. C) Date of diagnosis of the cases identified in the two sources.
(PDF)

**S1 Table. Biobank fields and variable definitions.**
(PDF)

**S2 Table. Allele counts at the lead SNPs from GWAS1[4], GWAS2[5], and GWAS3[6] in all UK Biobank participants using the imputed UK Biobank datasets.**
(PDF)

**S3 Table. Counts of genotype combinations at the lead SNPs from GWAS1[4] and GWAS2[5] in all UK Biobank participants using both the imputed UK Biobank datasets.**
(PDF)

**S4 Table. Phenotype per combined genotype group at HLADQA1 (rs2187668, test allele T) and PLA2R1 (rs4664308, test allele A) from GWAS1[4] in all UK Biobank participants using the directly genotyped UK Biobank dataset.**
(PDF)

**S5 Table. Phenotype and genotype by degree of proteinuria for all UK Biobank participants, with genotypes obtained from the imputed UK Biobank dataset at the lead SNPs from GWAS1[4] and GWAS2[5].**
(PDF)

**S6 Table. Environmental exposures–work-related.**
(PDF)

## Author Contributions

**Conceptualization:** Paul Brenchley.

**Data curation:** Stephen A. Roberts, Durga Kanigicherla.

**Formal analysis:** Kieran Blaikie, Stephen A. Roberts, Matthew Gittins, Mallory L. Downie, Sanjana Gupta, Catalin Voinescu, Horia Stanescu.

**Funding acquisition:** Paul Brenchley.

**Investigation:** Patrick Hamilton, Stephen A. Roberts, Durga Kanigicherla, Paul Brenchley.

**Methodology:** Patrick Hamilton, Stephen A. Roberts, Durga Kanigicherla, Robert Kleta, Paul Brenchley.

**Project administration:** Paul Brenchley.

**Supervision:** Robert Kleta, Paul Brenchley.

**Writing – original draft:** Patrick Hamilton.

**Writing – review & editing:** Kieran Blaikie, Stephen A. Roberts, Matthew Gittins, Mallory L. Downie, Sanjana Gupta, Catalin Voinescu, Durga Kanigicherla, Horia Stanescu, Robert Kleta, Paul Brenchley.

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
