## [Decision Letter · Decision Letter 0]

14 Mar 2023

PONE-D-23-02797Membranous Nephropathy in the UK BiobankPLOS ONE

Dear Dr. Hamilton,

Thank you for submitting your manuscript to PLOS ONE. After careful consideration, we feel that it has merit but does not fully meet PLOS ONE’s publication criteria as it currently stands. Therefore, we invite you to submit a revised version of the manuscript that addresses the points raised during the review process.

We look forward to receiving your revised manuscript.

Kind regards,

Ali M Shendi 

Academic Editor

PLOS ONE

Journal Requirements:

"We acknowledge funding from Kidney Research UK (KRUK) for the Stoneygate Foundation Grant JFS_IN_003_20160914

MLD was supported by the KRESCENT post-doctoral fellowship from the Kidney Foundation of Canada"

Additional Editor Comments:

-It seems that the authors meant expressing different albuminuria grades in mg/mmol (rather than mg/g). Please revise the unit used and corresponding values in the text and table 3, S4, S5.

-In the last line in table 6 (distance from main road "per 100 meters"), it is not clear what the figures represent.

Reviewers' comments:

Reviewer's Responses to Questions

**Comments to the Author**

1. Is the manuscript technically sound, and do the data support the conclusions?

Reviewer #1: Yes

Reviewer #2: Yes

Reviewer #3: Yes

2. Has the statistical analysis been performed appropriately and rigorously? 

Reviewer #1: Yes

Reviewer #2: Yes

Reviewer #3: I Don't Know

3. Have the authors made all data underlying the findings in their manuscript fully available?

Reviewer #1: Yes

Reviewer #2: Yes

Reviewer #3: Yes

4. Is the manuscript presented in an intelligible fashion and written in standard English?

Reviewer #1: Yes

Reviewer #2: Yes

Reviewer #3: Yes

5. Review Comments to the Author

Reviewer 1: 

The authors did an excellent work, using the UK BioBank data to identify patients with confirmed or potential MN according to genetic predisposition. It would be great if the authors could build on their work, by following up patients with putative MN for development of antiPLA2R Ab, and full clinical picture, and see whether this can be prevented earlier or not. May be an annual check for proteinuria, eGFR and serum antiPLA2R Ab level sounds reasonable.

Please consider adjusting these changes to make the manuscript more understandable to the nephrology community:

1. Please use kidney instead of renal whenever possible, as per KDIGO latest recommendation. ESKD instead of ESRD. kidney functions instead of renal function, and so forth ...

2. There is a repeated error using albuminemia instead of albuminuria in lines 173, 209, 283.

3. Line 293: there is a symbol of question mark, what is it?

4. Please consider using the KDIGO grading of proteinuria for better understand of the degree of protienuria. uACR less than 30 is normal, and not considered proteinuria at all, A1 (normal) uACR less than 30, A2 (moderately increased) uACR 30-300 mg/g, A3 (severely increased) uACR <300 mg/g and avoid using micro and macro albuminuria terms.

Reviewer 2: 

Well written and presented.

Several limitations which the authors acknowledge - small numbers, inherent bias of the UK biobank population.

Important study though how to integrate this additional information into clinical practice is not clear.

Reviewer 3: 

No major or critical comments. The article is well wrriten. The methodology is total clear with a very good number of subjects involved. The discussion is also well written with clear details.

6. PLOS authors have the option to publish the peer review history of their article (what does this mean?). If published, this will include your full peer review and any attached files.

Reviewer #1: No

Reviewer #2: No

Reviewer #3: No

---

## [Author Response · Author response to Decision Letter 0]

3 Apr 2023

PLOS ONE Journal Requirements:

Manuscript updated to reflect journals formatting guidelines.

No specific ethical approval is required. The UK Biobank has approval from the North West Multi-centre Research Ethics Committee (MREC) as a Research Tissue Bank (RTB). This approval means researchers do not require separate ethical clearance and can operate under the RTB approval (there are certain exceptions to this which are set out in the Access Procedures, such as re-contact applications).

This RTB approval was granted initially in 2011 and it is renewal on a 5-yearly cycle: hence UK Biobank successfully applied to renew it in 2016 and 2021. UK Biobank will in due course apply for renewal effective in 2026. These renewal applications and approvals are shown on the website.

https://www.ukbiobank.ac.uk/learn-more-about-uk-biobank/about-us/ethics

Methods section updated to state:

‘Ethics statement

The UK Biobank has ethical approval from the North West Multi-centre Research Ethics Committee (MREC reference 21/NW/0157) as a Research Tissue Bank (RTB) and therefore no separate ethical approval was required.’

"We acknowledge funding from Kidney Research UK (KRUK) for the Stoneygate Foundation Grant JFS_IN_003_20160914

MLD was supported by the KRESCENT post-doctoral fellowship from the Kidney Foundation of Canada"

Acknowledgements section updated to state:

‘Acknowledgements/Funding statement

This was an approved study (I.D. 1618) by UK Biobank (http:/ukbiobank.org). We acknowledge funding from Kidney Research UK (KRUK) for the Stoneygate Foundation Grant JFS_IN_003_20160914

MLD was supported by the KRESCENT post-doctoral fellowship from the Kidney Foundation of Canada

There was no additional external funding received for this study’

All data is from the UK biobank and therefore not permitted to be shared publicly. However, the data can be accessed by applying directly to the UK Biobank by any researcher worldwide through its standard application process - https://www.ukbiobank.ac.uk/enable-your-research/apply-for-access

Methods section updated to state:

‘Data availability statement

All data used in this study has been obtained from the UK Biobank and due to legal constraints, the data is not permitted to be shared. All data can be accessed from UK Biobank through its standard application process’

Ethics statement included in Methods section

References reviewed.

Additional Editor Comments:

-It seems that the authors meant expressing different albuminuria grades in mg/mmol (rather than mg/g). Please revise the unit used and corresponding values in the text and table 3, S4, S5.

Many thanks, we have updated the text and tables accordingly.

-In the last line in table 6 (distance from main road "per 100 meters"), it is not clear what the figures represent.

This represents the distance from Home to the main road (manuscript now updated to state this).

Reviewer 1: 

The authors did an excellent work, using the UK BioBank data to identify patients with confirmed or potential MN according to genetic predisposition. It would be great if the authors could build on their work, by following up patients with putative MN for development of antiPLA2R Ab, and full clinical picture, and see whether this can be prevented earlier or not. May be an annual check for proteinuria, eGFR and serum antiPLA2R Ab level sounds reasonable. Please consider adjusting these changes to make the manuscript more understandable to the nephrology community:

1. Please use kidney instead of renal whenever possible, as per KDIGO latest recommendation. ESKD instead of ESRD. kidney functions instead of renal function, and so forth ...

Many thanks for this, we have now updated the manuscript with renal changed to kidney in all instances, and ESRD changed to ESKD. 

2. There is a repeated error using albuminemia instead of albuminuria in lines 173, 209, 283.

Again, many thanks for identifying this error, all instances have now been changed to albuminuria. 

3. Line 293: there is a symbol of question mark, what is it?

Many thanks for identifying this, the ‘?’ is actually ‘�’ and the manuscript has now been updated to reflect this.

4. Please consider using the KDIGO grading of proteinuria for better understand of the degree of proteinuria. uACR less than 30 is normal, and not considered proteinuria at all, A1 (normal) uACR less than 30, A2 (moderately increased) uACR 30-300 mg/g, A3 (severely increased) uACR <300 mg/g and avoid using micro and macro albuminuria terms.

We have also updated the manuscript to reflect this as well. Many thanks.

---

## [Editor Report · Decision Letter 1]

10 Apr 2023

Membranous Nephropathy in the UK Biobank

PONE-D-23-02797R1

Dear Dr. Hamilton,

We’re pleased to inform you that your manuscript has been judged scientifically suitable for publication and will be formally accepted for publication once it meets all outstanding technical requirements.

Kind regards,

Ali M Shendi

Academic Editor

PLOS ONE
---

## [Editor Report · Acceptance letter]

19 Apr 2023

PONE-D-23-02797R1 

Membranous Nephropathy in the UK Biobank 

Dear Dr. Hamilton:

I'm pleased to inform you that your manuscript has been deemed suitable for publication in PLOS ONE. Congratulations! Your manuscript is now with our production department. 

Kind regards, 

on behalf of

Dr. Ali Moustafa Shendi Mohamed 

Academic Editor

PLOS ONE